



# Applications of Machine Learning and Artificial Intelligence in Tropospheric Ozone Research

Sebastian H. M. Hickman[1,*], Makoto M. Kelp[2,*], Paul T. Griffiths[3,*], Kelsey Doerksen[4,*], Kazuyuki Miyazaki[5,*], Elyse A. Pennington[5,*], Gerbrand Koren[6,*], Fernando Iglesias-Suarez[7,*], Martin G. Schultz[8,*], Kai-Lan Chang[9,10], Owen R. Cooper[9], Alex Archibald[3], Roberto Sommariva[11,12], David Carlson[13], Hantao Wang[14], J. Jason West[14], and Zhenze Liu[15]

[1]Yusuf Hamied Department of Chemistry, University of Cambridge, Cambridge, UK
[2]Doerr School of Sustainability, Stanford University, CA, USA
[3]National Centre for Atmospheric Science, Cambridge University
[4]Department of Computer Science, University of Oxford
[5]Jet Propulsion Laboratory, California Institute of Technology, Pasadena, CA, USA
[6]Copernicus Institute of Sustainable Development, Utrecht University, Utrecht, the Netherlands
[7]Predictia Intelligent Data Solutions S.L., Santander, Spain
[8]Jülich Supercomputing Centre, Forschungszentrum Jülich, Jülich, Germany
[9]NOAA Chemical Sciences Laboratory, Boulder, CO, USA
[10]Cooperative Institute for Research in Environmental Sciences, University of Colorado Boulder,
[11] School of Geography, Earth and Environmental Sciences, University of Birmingham, Birmingham, UK
[12]School of Chemistry, University of Leicester, Leicester, UK
[13]Civil and Environmental Engineering, Duke University, NC, USA
[14]Department of Environmental Sciences and Engineering, University of North Carolina, Chapel Hill, NC, USA
[15]Jiangsu Key Laboratory of Atmospheric Environment Monitoring and Pollution Control, School of Environmental Science and Engineering, Nanjing University of Information Science and Technology, Nanjing, China
[*]These authors contributed equally to this work.

**Correspondence:** Paul Griffiths (paul.griffiths@ncas.ac.uk)

**Abstract.**

Machine learning (ML) is transforming atmospheric chemistry, offering powerful tools to address challenges in tropospheric ozone research, a critical area for climate resilience and public health. As in adjacent fields, ML approaches complement existing research by learning patterns from ever-increasing volumes of atmospheric and environmental data relevant to ozone.

5 We highlight the rapid progress made in the field since Phase 1 of the Tropospheric Ozone Assessment Report, focussing particularly on the most active areas of research, namely short-term ozone forecasting, emulation of atmospheric chemistry and the use of remote sensing for ozone estimation. Despite these advances, many challenges in the field remain, including the quality of data, benchmarks, and limited model generalisation and explainability. This review provides a comprehensive synthesis of recent advancements, highlights critical challenges, and proposes actionable pathways to further advance ML

10 applications in ozone research. Achieving this potential will require close collaborations across atmospheric chemistry, ML and computational science, aimed at addressing key challenges such as the development of global benchmark datasets and robust, explainable models.



## 1 Introduction

Tropospheric ozone is a harmful atmospheric pollutant and an important greenhouse gas, contributing to both environmental
and public health issues. Long-term exposure to elevated ozone levels is linked to hundreds of thousands of premature deaths
globally each year (Malashock et al., 2022; Malley et al., 2017; Health Effects Institute, 2024). Short-term exposure can cause
serious negative health impacts (Bell et al., 2014) including reduced lung function, particularly in individuals with pre-existing
medical conditions (EPA, 2020). Beyond its health impacts, tropospheric ozone significantly damages vegetation in natural
ecosystems and agricultural fields (Mills et al., 2018) and can act as a climate forcer in the upper troposphere. In addition,
ozone plays a critical role in tropospheric chemistry, both as a source of oxidants and as a primary oxidant itself (Monks et al.,
2015).

Ozone is not directly emitted into the troposphere but is photochemically produced in the presence of sunlight by reactions
involving its precursor gases: carbon monoxide (CO), methane ($CH_4$), volatile organic compounds (VOCs), and nitrogen
oxides $NO_X$ ($NO_X$, $NO+NO_2$). In addition, ozone is transported from the stratosphere into the troposphere. The removal
of tropospheric ozone is controlled by chemical loss and deposition to the surface (Archibald et al., 2020). The lifetime of
ozone in the troposphere ranges from days to weeks, depending on local chemical and meteorological conditions (Lelieveld
and Dentener, 2000; Monks et al., 2015). This variability allows ozone and its precursors to be transported over long distances
from their sources (Fiore et al., 2009).

Modeling tropospheric ozone is therefore a challenging task due to the complex coupling of these chemical and physical
processes that control its local concentrations across different spatial and temporal scales, as detailed in Figure 1. Traditionally,
concentrations of ozone and other chemical species are calculated using numerical models of the atmosphere that represent
these processes across a wide range of spatial scales, from high-resolution urban models (meter-scale) to global chemistry-
climate models with resolutions ranging from tens to hundreds of kilometers (e.g. Morgenstern et al. (2017)).

Despite the success of ozone simulations in air quality and climate research, there are still challenges to reach accurate
simulations of ozone (Young et al., 2018). For example, although ozone is the longest- and most-measured trace gas in the
observational record, large uncertainties still exist in global model estimates of tropospheric ozone and its trends. Observations
from ground stations, ozonesondes, and satellites indicate that tropospheric ozone has generally increased in recent decades
(Ziemke et al., 2019; Young et al., 2018; Gulev et al., 2021). While global atmospheric chemistry models agree that the global
tropospheric ozone burden has increased from pre-industrial times to the present day, they vary regarding the spatial distribution
and magnitude of the increase (Skeie et al., 2020; Christiansen et al., 2022; Fiore et al., 2022). Potential sources driving this
model bias include uncertainties in tropical emissions (Zhang et al., 2021), nonlinear $NO_X$-VOC chemistry (Shah et al., 2023),
stratosphere-troposphere exchange (Neu et al., 2014), boundary layer mixing (Lu et al., 2019), missing chemical mechanisms
such as halogen chemistry (Wang et al., 2015), and deposition (Clifton et al., 2020).

The variation of ozone at various scales is shown in Figure 2. The figure shows the diurnal and annual cycles of ozone at four
sites from the TOAR database: Mauna Loa Observatory, a Pacific mountain station, based in Hawaii, USA; Minamitorishima,
a Pacific island station in Japan; a regional continental background site, Borken, Germany, and an urban, roadside site, Maryle-



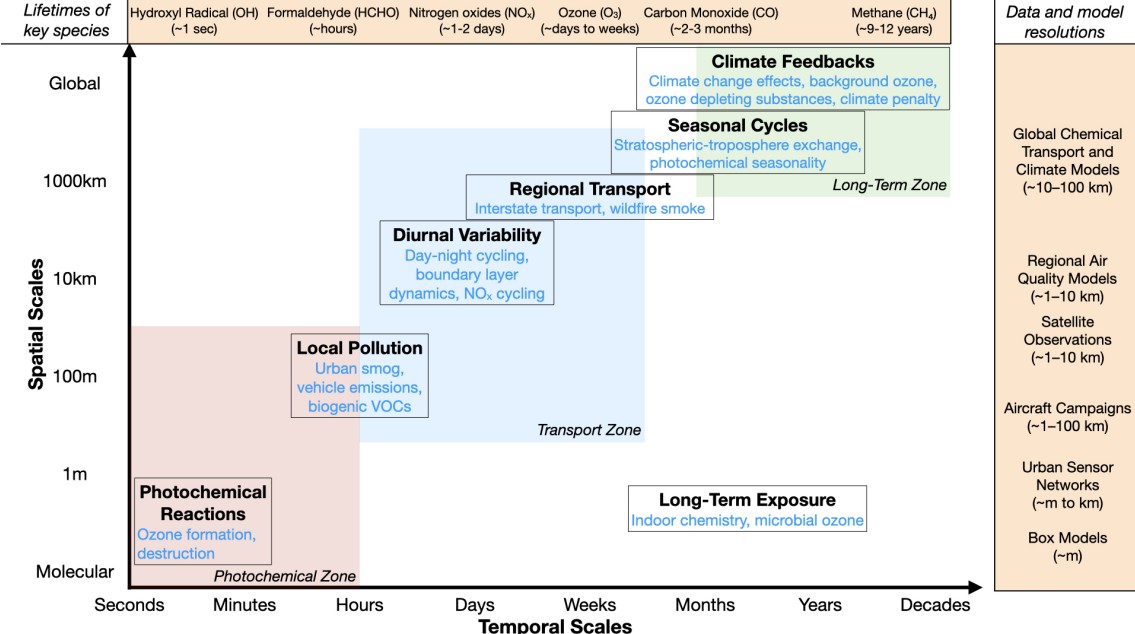

**Figure 1.** Spatial and temporal scales of tropospheric ozone chemistry processes. The x-axis shows timescales, from rapid photochemical reactions to long-term climate feedbacks, and the y-axis shows spatial scales, from local pollution to global atmospheric transport. Species lifetimes and relevant data sources and models are displayed to illustrate the range and scales of phenomena and methods used to study ozone chemistry.

bone Road, London, UK. There is little consistency between the diurnal cycles at the various sites: the remote Pacific site sees little diurnal variation in ozone, but a strong seasonal cycle, with levels reaching a minimum in the summer. In contrast, the continental, rural background site in Germany has a strong diurnal cycle, peaking in the late afternoon, and a strong seasonal cycle with a summertime maximum. The long-term trends in ozone, although weak, also vary between the sites, with both modest increases (London Marylebone Road) and decreases (Minamitorishima) being seen. The lower panel shows variation in ozone trends across the globe, ranging between -3 and 3 ppbv per year, across remote sites.

Machine learning (ML) approaches, which can learn and reproduce nonlinear characteristics of a system from data (Hornik et al., 1989), may provide a valuable complement to physical models. As the quantity and quality of observational data on ozone (Schultz et al., 2017) and on the broader Earth system (Agapiou, 2017; Reichstein et al., 2019) continue to grow, ML is becoming an increasingly viable tool for advancing ozone research.

Figure 3 highlights progress in the field of ML as applied to weather and climate science, which has been rapid since the publication of the first phase of the TOAR assessment. In their review of the state of the field of weather forecasting, Bauer et al. (2015) note many areas of progress for the field, including model throughput, the process-level detail of then current models, and the use of data assimilation techniques to improve the fidelity of the model's initial state. The impact of ML methods was not anticipated. Rasp et al. (2018) demonstrated the potential for Deep Learning techniques to augment existing





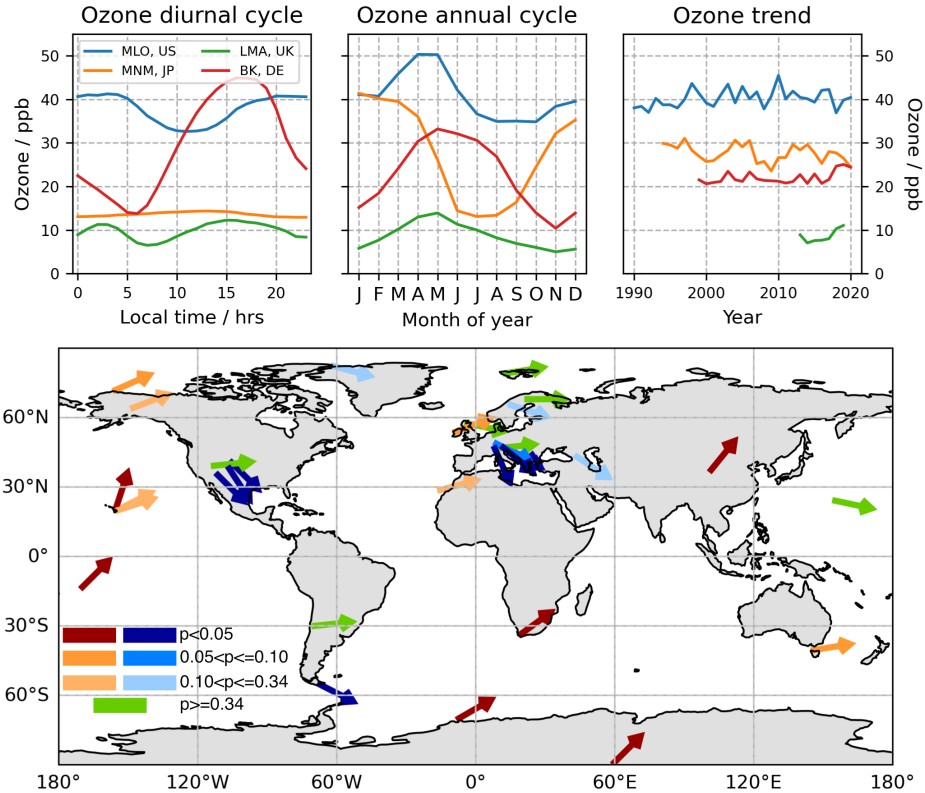

**Figure 2.** Upper panel: data from the TOAR ozone database for four sites in the northern hemisphere, showing diurnal and seasonal cycles in ozone, and the long-term ozone trend. Lower panel: Long-term ozone trends based on monthly anomalies at remote surface sites. Data from Cooper et al. (2020) and replotted here

models in providing an alternative, complementary and physically consistent description of sub-grid scale processes, such as cloud microphysics. The coupling of a fast, accurate, data-driven module, trained on finer scale simulations, to a larger scale host climate model exemplified one of the potential ways that ML approaches can contribute to the improvement of climate

and weather models. Subsequent studies have shown in various ways the advantages of ML over traditional numerical models, particularly in terms of computational efficiency and in the ability to learn from large datasets, as demonstrated by the success of data-driven nowcasting and weather forecasting models (Bi et al., 2023; Lam et al., 2023; Price et al., 2024).

While most ozone ML studies to date have prioritized prediction, ML also has great potential to enhance process-level understanding. Observational data, when integrated with model simulations through data assimilation techniques, have already

improved the understanding of emissions and atmospheric chemistry by reducing uncertainties (Miyazaki et al., 2020b). ML can complement these efforts by combining observational data with model outputs, emulating model components, or enabling computationally cheaper simulations, thereby efficiently diagnosing sources of error in global atmospheric models and improving tropospheric ozone estimates. However, ML also has limitations, such as challenges in generalization, validation, and





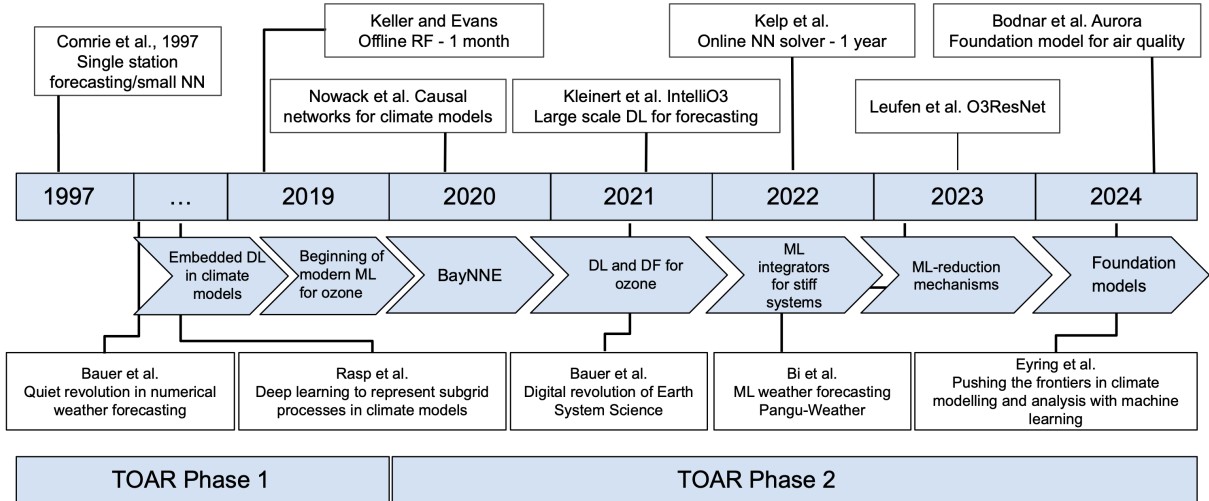

**Figure 3.** Timeline of a selection of studies using ML in ozone research, aligned with a selection of papers using ML in wider weather and climate modeling research. In both wider Earth system modeling research, and in ozone research there has been rapid progress over the last five years, as noted by landmark review papers highlighted in the Figure.

interpretability. Addressing these issues may be particularly relevant for the ozone modeling community where both predictive

accuracy and physical understanding are valued.

In this Perspective, we provide an overview of the state of ML in tropospheric ozone research, review previous applications of ML to various problems related to ozone, and discuss persistent challenges and emerging opportunities. We detail the bottlenecks to progress in applying ML to ozone problems, including the limitations of available training data, heterogeneity of data sources, and computational limits. We suggest that the use of ML can provide tangible improvement over existing methods

in some aspects of ozone research and we provide possible paths towards further improvement to complement existing models and initiatives.

We highlight three areas where ML for ozone has been most widely applied: forecasts based on ground-based observations are reviewed in Section 2, methods for complementing or replacing parameterizations in numerical models of atmospheric chemistry and transport are discussed in Section 3, and ML models that are using satellite data or combined data products

are presented in section 4. Limitations specific to each section are briefly mentioned, and we find common limitations across domains. Section 5 highlights and further details these cross-cutting issues and limitations with the application of ML to ozone studies, while Section 6 describes future directions for the field, highlighting emerging approaches that seek to address the cross-cutting challenges.





## 2 Applications of ML to in-situ ozone observations: short-term ground level ozone forecasting

### 2.1 Background


State-of-the-art air quality forecasts, typically on the timescales of hours or days and up to a few days ahead, are based on the output of numerical chemical transport models (Marécal et al., 2015). These models may be run at higher spatial resolution for the area of interest (Savage et al., 2013), in order to better represent processes controlling air pollution at the local level, and may be post-processed to more accurately represent observations (Casciaro et al., 2022). As with other air pollutants, notably

$PM_{2.5}$ (Feng et al., 2015), ML is increasingly being directly applied to the task of short-term, ground-level ozone forecasting, and to bias-correct existing air quality forecasting systems with considerable success. The availability of large and growing observational datasets has facilitated these advances (Schultz et al., 2017). However, forecasting ozone concentrations as time series with ML comes with significant challenges: forecasting ozone is a spatiotemporal problem, and ozone is controlled by processes of varying spatial and temporal scales as shown in Figure 1.

The short-term forecasting of air pollutants including ozone, i.e. predictions of expected concentrations over 1-4 days, is relevant for public health and scientific questions. One important task is to provide the public with accurate and timely warnings of poor air quality, which may influence public behavior and encourage precautions, leading to reduced exposure to poor air quality (Buonocore et al., 2021; Hahm and Yoon, 2021; Alari et al., 2021; Saberian et al., 2017). Existing ML forecast methods are well suited to this task. Similar ML architectures and workflows have also been used to fill in missing data in

observational datasets (Betancourt et al., 2022; Wu et al., 2024).

Many short-term forecasting studies using ML have focused on forecasting only at selected observational stations, using observed ozone and additional chemical species, and meteorological variables where they are available from individual stations or external datasets (Comrie, 1997; Cobourn et al., 2000; Kolehmainen et al., 2001; Eslami et al., 2020; Sayeed et al., 2021; Leufen et al., 2023; Hickman et al., 2023). Furthermore, since it is difficult to downscale a relatively coarse chemical transport

model (CTM) at specific locations, using time series data from a particular station is an attractive way to make predictions at particular locations. However, approaches of this kind do not necessarily provide ozone forecasts across all locations that may be of interest, as a gridded model product model might.

### 2.2 Progress and State of the Science

As with other fields, the advances in ML-based ozone forecasting have been pushed by developments on two axes - first,

increasing quantities of data and second, larger models with more appropriate inductive biases. The field has a long history (see Figure 3), with studies being published even during the most recent artificial intelligence (AI) "winter", beginning with a feed-forward neural network (NN) in 1996 (Yi and Prybutok, 1996). Comrie (1997) illustrated that a neural network (NN) could be used to forecast ozone at eight stations in the USA. This was followed by further feed-forward NN approaches, often with datasets drawn from a single location or city (Cobourn et al., 2000; Kolehmainen et al., 2001). Neural methods were typically

evaluated in comparison with (autoregressive) regression models, often finding that NNs were better able to forecast ozone concentrations and extrema on test data (Nunnari et al., 1998; Schlink et al., 2003; Chaloulakou et al., 2003), although the




improvement was often only marginal. Alongside the successes of feed-forward NN architectures, other work drew attention to methods seen to be more interpretable, such as fuzzy logic systems and regression trees (Gardner and Dorling, 2000; Heo and Kim, 2004). Further work leveraged methodological advances in ML architectures designed for temporal data, including the use of recurrent neural networks (RNNs) and convolutional neural networks (CNNs) to account for lagged relationships in the time series data (Eslami et al., 2020; Sayeed et al., 2021; Kleinert et al., 2021). Recent work has combined architectures to model the relationships that control ozone, including combining components such as transformers and CNNs to account for the temporal and spatial information relevant to forecasting (Chen et al., 2022; Cheng et al., 2022; Han et al., 2023). However, datasets have typically been limited to single countries or cities, due to the lack of a combined database of station measurements. The introduction of the Tropospheric Ozone Assessment Report (TOAR) surface database (Schultz et al., 2017) and the TOAR-II database have facilitated recent studies on data drawn from multiple countries (Leufen et al., 2023; Hickman et al., 2023). The importance of curation of large datasets for scientific progress in ML is highlighted below in the Outlook section.

Increasingly, more complex architectures are being used to enhance the accuracy of ozone forecasts, and more data are being included as input to the models. The inputs that are relevant to the physical drivers of ozone concentrations, such as past observations of ozone and covariates, and nearby covariates, reflect processes that control ozone observations, and feasibly contribute to improved ozone forecasting and infilling. Recently, methods on the scale of the ML architectures and data used for weather forecasting (Bi et al., 2023; Lam et al., 2023) have been transferred to ozone forecasting by leveraging very large datasets and models. In weather studies there is work on forecasting at observation stations using these methods, and transferring these methods to forecast ozone at ground-level stations is feasible (Manshausen et al., 2024).

General trends have emerged from recent forecasting studies with ML. Typically, models trained on many sites perform better than models trained to predict ozone at a single station, and larger deeper models are better able to forecast given sufficient data. However, it is difficult to determine precisely which individual features of models are contributing most to improved accuracy as studies typically use different test data. The establishment of a robust benchmark dataset, which could be used across the community to evaluate the performance of different methodologies would allow rigorous comparison between competing methods. A dataset of this kind for weather forecasting, WeatherBench, and its successor (Rasp et al., 2020, 2024) continue to facilitate effective inter-comparison between different ML and numerical models. A similar dataset to perform the same function for ozone forecasting would allow the robust comparison of different methods, and may guide the field towards more accurate models. This benchmark dataset would need to cover a range of environments, including extreme events. Careful curation and data fusion of the TOAR surface ozone database with other relevant datasets might provide a robust and representative benchmark dataset, building on existing work (Betancourt et al., 2021).

## 2.3 Future Outlook

While forecasting ground-level ozone has seen some successes, there are still some necessary steps before we have reliable operational ML forecasts. As in other fields, the community has seen improved performance with scaling datasets, and training across a variety of environments with rich data (Leufen et al., 2023). While the TOAR database of surface observations provides





a large collection of in-situ data, the community is still missing a fused, global and gridded benchmark dataset that could be used to robustly compare methods (Dueben et al., 2022; Rasp et al., 2024), and explore model deficiencies. Creating this dataset should be a collaborative effort that may provide the step-change in ground level air pollution forecasting that other benchmark datasets have precipitated in other fields (Deng et al., 2009). Part of this benchmark should focus on extreme events
and probabilistic metrics, as currently the community does not evaluate performance on extremes thoroughly, and encouraging probabilistic models and metrics may improve performance on extremes. While generalization is less of an issue for short-term forecasting, building models that can generalize to undersampled regions of the world would facilitate better pollution forecasting in countries with fewer data. Finally, work on interpretability and explainability is limited, and further study may give more insight into the internal working of these models, and possibly insight into physical processes.

## 3   Applications of ML methods in atmospheric chemical modeling

### 3.1   Background

Global modeling of atmospheric chemistry is a grand computational challenge due to the high dimensionality of coupled chemical species, the nonlinearity and numerical stiffness of solving chemical mechanisms, and interactions with transport on all scales. The inclusion of comprehensive atmospheric chemistry in Earth system models (ESMs)—which simulate the
interactions between the atmosphere, oceans, land surface, and biosphere—is a priority science frontier (National Research Council (U.S.)., 2012). Atmospheric chemistry in these models is typically represented by chemical transport models (CTMs), which focus on the distribution and chemical evolution of species in the atmosphere. For some applications, chemistry-climate models (CCMs) may further couple chemical processes with climate dynamics, allowing feedback between chemistry and climate. Current atmospheric chemistry models integrate the coupled chemical kinetic equations for mechanism species over
model time steps using high-order implicit numerical solvers, but these solvers are computationally expensive (Sandu et al., 1997) and often dominate the cost of an atmospheric simulation (Eastham et al., 2018). Such costs put the inclusion of atmospheric chemistry in tension with other computationally intensive ESM/CCM priorities such as increased spatial resolution or ensemble simulations. The current slowdown in the rate of increase in the speed of computer CPUs—the "end of Moore's law"—underscores the need for computationally efficient approaches (Theis and Wong, 2017).
Chemical solvers in atmospheric models compute the local evolution of species concentrations over a chemical time step that may range from minutes to hours depending on the model Brasseur and Jacob (2017). The chemical mechanisms used in global/regional atmospheric models and ESMs typically include $\mathcal{O}(100)$ coupled species and is a multiscale system with chemical lifetimes ranging from less than a second to much larger than the model time step. High-order implicit solvers can integrate this system of stiff coupled differential equations with high accuracy, and fast implementations of these schemes are
available, but they are still extremely costly for atmospheric models. Atmospheric models may combat that cost by decreasing the size of the chemical mechanism, breaking down the stiffness of the problem, or using lower-order approximations. However, these methods rarely achieve a speedup of more than a factor of two (Lin et al., 2023; Shen et al., 2020) and sometimes lead to loss of accuracy in the model results. As a consequence, these computational barriers limit the ability for high-resolution





simulations, prevent detailed uncertainty analyses, and complicate the coupling of atmospheric chemistry into CCMs/ESMs
for long-term climate simulations without significant compute resources.

ML methods could be transformative in this area for both reducing the cost of an atmospheric chemistry simulation and
facilitating their incorporation into ESMs. ML methods seem well-suited to replace chemical solvers in atmospheric models
because the chemical computation is very repetitive, involving the integration of similar conditions in neighboring grid cells
and successive time steps. However, the large number of coupled species brings a 'curse of dimensionality' to the problem, and
ML methods have no check on error growth, unlike in standard chemical solvers where errors are dampened by the negative
response to perturbations (Le Chatelier's principle). Additionally, while ML models are commonly trained and deployed using
software libraries written in Python, the successful integration of ML-based algorithms and parameterizations within 3-D CTMs
remains limited. A potential barrier is the difficulty of calling Python libraries from Fortran-based models, which Fortran cannot
do so natively. However, recent advances in ML software can package ML code into static or dynamic libraries for flexible
deployment.

There are a variety of examples using output from CTM and air quality models in tandem with satellite measurements,
ground-level monitoring observations, meteorological reanalysis data, and emissions databases to predict or improve forecasts
of ozone. These improvements can be achieved through various methods, including data fusion techniques and the creation of
reduced-order models. However, because such studies involve additional data besides model outputs, they will be discussed in
other sections (Sections 2 and 4). Here, we emphasize studies primarily employing data from atmospheric chemical models.

## 3.2   Progress and State of the Science

Largely, ML methods in atmospheric chemistry modeling currently involve emulating model components to improve model
parameterizations, reduce computational bottlenecks, and create simplified, reduced-order models. Here, emulation refers to
an ML model reproducing the same calculations as a component of a complex physical or simulated system for a set of inputs.
There exists a growing number of studies forecasting ozone on short- (hourly) (Yafouz et al., 2022) and longer-term (Du et al.,
2022; Chen et al., 2023) timescales, spanning from city- (Ojha et al., 2021) to regional-level (Ortiz et al., 2021) spatial scales.
However, few of these studies have been implemented in operational settings (i.e., within CTMs, CCMs) and offer insight
beyond that of traditional model-to-observation comparison methods.

### 3.2.1   ML emulation and reduced order modeling

Xing et al. (2020) used a hierarchy of ML models containing a CNN and long short-term memory (LSTM) network to predict
ozone concentrations from CMAQ model output over 7-day forecast periods. Kuo and Fu (2023) investigated how accurately
ML models can learn the ozone-$NO_X$-VOC chemical relationships in a chemical mechanism and found that their ML model
produced distorted $NO_X$ and VOC-limited isopleths when only trained on CMAQ model outputs. Kelp et al. (2022) trained an
NN integrator in a photochemical box model, including an encoder/decoder to decrease dimensionality, and a recursive feed-
back loop over 24-hr integration time to control error growth. They found that they could compress the 101-species dimension
of their mechanism into 16 features without significant error penalty and avoid error growth within a selected time horizon,



though error increases beyond this window. Yang et al. (2024) created an ML surrogate for a low-dimensional (11 species) chemical box model that both compresses the dimensionality of the chemical mechanism and reduces the numerical stiffness of the problem. They achieve numerical stability within a 9-day training window but acknowledge that such an approach may be difficult for more complex and higher-dimensional chemical mechanisms. Liu et al. (2024) employed a Fourier Neural Operator with time-embedded attention to calculate chemical concentration changes as a learnable time-dependent process. They achieved higher accuracy metrics compared to standard neural operators and U-Nets in simple box model-like simulations.

### 3.2.2 ML models implemented within global CTMs

While there is a growing literature on using ML to emulate and improve the representation of atmospheric processes, few have implemented these ML models online within CTMs/ESMs to evaluate their effectiveness. Keller and Evans (2019) created a random forest integrator for the GEOS-Chem global 3-D CTM driven by re-analyzed meteorological data. They achieved successful short-term simulations but found large error growth after a few weeks. Liu et al. (2022a) developed a gas-phase NN solver for the CMAQ regional CTM over China, combining a standard implicit solver for radicals and oxidants with an ML solver for VOCs. They achieved an order of magnitude speedup over a 1-month simulation but with error growth over remote ocean grid cells. Shen et al. (2022) used an unsupervised ML algorithm (simulated annealing) to create submechanisms of the full chemical mechanism in GEOS-Chem for which they solve the coupled kinetic system only for the fast species in the submechanism. The computational cost of the chemical integration decreased by 50% and the relative difference in ozone was <0.5% in the troposphere and <0.1% in the stratosphere over 8-year simulations. Kelp et al. (2022) implemented the low-dimensional "Super-Fast" chemical mechanism in GEOS-Chem using online training of the ML emulator, achieving stable 1-year simulations for ozone prediction with less than 10% bias compared to the reference and reducing computational cost by a factor of five. However, their ML solver had relatively lower accuracy in pristine marine regimes with lower chemical concentrations. Xia et al. (2024) implemented a self-attention transformer chemical solver online into the WRF-Chem CTM achieving an eight-time speedup over the conventional solver with an average NMB of 2.63% for 74 species. Their approach shows promise for accurate predictions of chemical concentrations with low overhead when coupling the ML solver to the CTM, but simulations were only run for 15 days and stability over longer time scales (>1 year) remains to be seen.

Few studies have implemented ML models directly within CTMs/ESMs to evaluate their effectiveness, although the necessary tools exist. While ML models are typically trained and deployed using Python libraries, integration of these models into CTMs remains limited because CTMs are written in Fortran, which cannot natively call Python. Current solutions include rewriting models in neural Fortran (Keller and Evans, 2019), using the C Foreign Function Interface (CFFI) to create C-style bindings for Python scripts (Kelp et al., 2022; Zhong et al., 2023), or packaging ML models as callable static or dynamic libraries using TorchScript and LibTorch (Xia et al., 2024). Depending on the architecture and complexity of the coupled ML model, all coupling methods result in a speedup over the conventional reference solver. See review by de Burgh-Day and Leeuwenburg (2023) for greater discussion on using ML for facilitating speedups and computational cost reductions in atmospheric science models.



### 3.2.3 ML modeling processes affecting ozone chemistry

A number of ML and data-driven advances have been made for CTM modeling that are separate from creating an ML chemical solver. Wiser et al. (2023) and Wang et al. (2023) created automated chemical mechanism reduction approaches to reduce the high dimensionality of the VOC precursors of ozone and secondary organic aerosol. Sturm and Wexler (2022, 2020) developed methods to enforce mass and stoichiometric conservation rules in outputs from ML emulators.Anderson et al. (2022) used gradient-boosted regression trees to develop a parametrization for the OH radical, a key driver of $O_3$ formation, for CCM models. Similarly, Zhu et al. (2022) trained an ML model on CTM output parameters and satellite observations from OMI to predict urban OH concentrations. Huang and Seinfeld (2022) created an NN-assisted Euler integrator to speed up the iterative computations within an implicit solver routine.

Silva et al. (2019) developed an ML parameterization for ozone dry deposition velocities using surface observations that outperformed those within CTMs for certain locations. Similarly, Ivatt and Evans (2020) created an XGBoost model trained on ozone surface observations and data from ozonesonde networks to predict and correct GEOS-Chem model biases. Liu et al. (2022a) developed a NN model to correct surface ozone in the UKESM model, finding that temperature drives biases over Northern Hemisphere continental areas while photolysis rates contribute to global ozone biases. To be sure, there is a growing literature on ML approaches for bias corrections on existing air quality modeling systems (Neal et al., 2014; Borrego et al., 2011; Silibello et al., 2015). These approaches generally learn the error between the output of a numerical model and some observations and then apply this error correction to the output of the numerical model. Nowack et al. (2018) used a hierarchy of ML methods to build temperature-based ozone parameterizations for climate model sensitivity simulations. Colombi et al. (2023) used random forests to remove the effect of weather coupled to ozone trends. Gouldsbrough et al. (2024) used a gradient-boosted tree to downscale ozone model output from the EMEP4UK CTM. Ye et al. (2022) used a random forest model to identify underlying causes of CTM bias in simulating daily surface ozone variability, finding that CTM underestimates in the dry deposition velocity and cloud optical depth on wet/cloudy days were the primary drivers over China.

Park et al. (2023) created a prototype ML discretization for a one-dimensional horizontal passive scalar advection, an operator component common to all CTMs, and achieved stability and orders of magnitude computational gain relative to the reference when coarse-grained. Sturm et al. (2023) developed a data-driven compression method for chemical tracers within a CTM and advected the compressed representation, achieving a computational gain of $1.5\times$ without loss of accuracy. There have been developments of ML emulators in box models for organic aerosol mechanisms detailing the ML models' accuracy with respect to interactions with ozone (Mouchel-Vallon and Hodzic, 2023; Schreck et al., 2022).

The photolysis frequencies used to inform ozone concentrations, calculated from the radiative transfer components of atmospheric models, can themselves be emulated using NNs (Lagerquist et al., 2021) and have the longest relative history of ML emulation for atmospheric modeling (Krasnopolsky et al., 2005, 2008).



### 3.3 Future Outlook and Priorities

The near-term future of integrating ML with 3-D atmospheric chemistry and climate modeling, particularly for tropospheric ozone prediction, relies on understanding the uncertainties and limitations of ML emulation. Such knowledge is essential for improving or approximating specific chemical parameterizations rather than attempting to replace full-scale, multiscale chem-
istry simulations. Key priorities include incorporating ML models into CTMs, CCMs, and ESMs, as well as characterizing their behavior over extended time scales (>1 year). While short-to-seasonal scale emulation may be suitable for forecasting horizons, it offers limited applicability for integrating comprehensive atmospheric chemistry into climate simulations. Additional priorities should include scaling regional ML models to a global context and identifying the strengths of ML (e.g., speed of operation, pattern recognition of high volumes of data) alongside its weaknesses (e.g., inability to approximate multiscale
dynamics, instability over time, resource-intensive training and black box nature).

In the context of using ML for model component emulation and replacing traditional components, a major challenge is managing coupled instabilities, where ML predictions accumulate errors and drift from the reference model over time. This instability is also prevalent in ML-based model emulation for weather and climate applications. Consequently, ML approaches are increasingly employed to develop end-to-end models that process raw input data (e.g., emissions, meteorological fields) and
directly predict outputs such as ozone concentrations. While end-to-end models bypass the challenges of emulating individual components and are less prone to short-term instabilities and operator splitting issues, they also limit the ability to track uncertainty metrics tied to physical parameters and processes.

Another emerging area is the use of Physics-Informed Neural Networks (PINNs), which embed physical laws into an ML model's architecture. While ML has been used to emulate traditional numerical solvers, this approach has often sacrificed accu-
racy and stability over long-term timescales in exchange for faster computations. Traditional solvers, though computationally expensive, do not exhibit these error instabilities (see Le Chatelier's principle). PINNs may offer a solution to balance speed with accuracy and have shown success in smaller-scale systems, such as single ODEs or PDEs (Karniadakis et al., 2021). Although their application to large groups of ODEs in atmospheric chemistry has not yet been widely explored, PINNs may hold the potential for achieving greater accuracy than traditional solvers, possibly overcoming some of the limitations of ML
approaches in atmospheric chemistry modeling. However, PINNs may not enforce physical laws perfectly and the inherent black-box nature of certain ML components limits their transparency compared to physical models, which provide clearer constraints. Therefore, while PINNs may address some computational challenges, their contribution to advancing process-level understanding in atmospheric chemistry remains uncertain.

At present, most ML models attempting to replace components of CTM/CCM parameterizations have used supervised
learning methods. These methods typically train using chemical data from box models or other solvers, aiming to faithfully reproduce outputs given initial conditions. In contrast, unsupervised learning, which identifies patterns or clusters in data without explicit labels, remains underexplored in this field. Greater emphasis is needed on understanding the factors influencing ML model performance, whether stemming from the ML model architecture, the quality and curation of training data, or uncertainties inherent in the physical processes of emissions, transport, and chemistry. To be sure, the advent of transformers





and Large Language Models (LLMs) offers a new frontier for ingesting and potentially improving AI modeling of atmospheric chemistry, but it will be crucial to understand the above points before pursuing even more complicated ML model architectures.

## 4 Applications of AI/ML methods to satellite observations

### 4.1 Background

Satellite measurements provide detailed information on the spatiotemporal distribution of atmospheric composition and related
parameters, such as those associated with surface air quality. Satellite measurements have greater spatial and temporal coverage compared to in-situ observations. Particularly in remote areas where in-situ observations are not available, they can fill the gaps in the sparse distribution.

Over the past few decades, multiple satellites have been launched to measure total ozone columns, including the Total Ozone Mapping Spectrometer (TOMS) (Prospero et al., 2002), Scanning Imaging Absorption Spectrometer for Atmospheric Cartog-
raphy (SCIAMACHY) (Bovensmann et al., 1999), Ozone Monitoring Instrument (OMI) (Veefkind et al., 2006), Global Ozone Monitoring Experiment (GOME) (Coldewey-Egbers et al., 2005), GOME-2 (Loyola et al., 2011) and the Sentinel-5P Tropospheric Monitoring Instrument (TROPOMI) (Garane et al., 2019). Geostationary satellite observations, such as the Tropospheric Emissions: Monitoring of Pollution (TEMPO) (Naeger, 2021) and Geostationary Environment Spectrometer (GEMS) (Baek et al., 2023) observe detailed spatial and temporal patterns of total column ozone and other air pollutants. However, total
column measurements cannot be used to provide insight into near-surface ozone because the amount of stratospheric ozone is much larger than the amount of tropospheric ozone.

Tropospheric ozone information has been directly retrieved using measurements from nadir-viewing thermal infrared (TIR) sounders, such as TES (Bowman et al., 2002) and IASI (Boynard et al., 2009), and by combining measurements from both ultraviolet (UV) and visible (VIS) wavelengths by TEMPO (Johnson et al., 2018). In addition, the limb-nadir matching method
employs stratospheric ozone data from limb-viewing measurements, such as those from the Microwave Limb Sounder (MLS), to derive tropospheric columns from observed total columns (Ziemke et al., 2019). Recently, multispectral satellite approaches, such as IAGI+GOME-2 (Cuesta et al., 2018) and TES+OMI (Colombi et al., 2021), have been implemented to derive tropospheric ozone profiles with increased sensitivity to the lower troposphere.

Nevertheless, satellite observations of ozone are still limited in spatial, temporal, and vertical resolution and are not suffi-
ciently sensitive to ground surface levels. On the other hand, measurements of precursors, such as $NO_2$ and $CH_2O$ from OMI, GOME-2, TROPOMI, and OMPS, have provided unprecedented information to assess the formation processes and surface concentrations of pollutants such as ozone and aerosols. Despite these advancements, technical challenges remain in accurately assessing near-surface air pollutant concentrations from satellite observations of precursors. AI/ML techniques can be used to fill the gaps in the information available from satellite observations and to improve the estimation of surface air pollutants.





### 4.2 Progress and State of the Science

AI/ML has been widely used in satellite applications, especially in remote sensing imagery (Maxwell et al., 2018) in the past and is becoming more widely applied to atmospheric composition data. AI/ML has been applied to satellite observations in two main categories: (1) to generate atmospheric concentration retrievals and blend multi-satellite products, and (2) to fill gaps in observational information, including surface concentrations and emissions estimates.

#### 4.2.1 ML models for fast retrievals and multi-satellite blending

Ozone retrieval is the task of estimating ozone profiles from spectrometers on satellites, which measure radiance spectra from the atmosphere. Usually carried out with numerical calculations based on physics, using ML for ozone retrieval is a burgeoning field. The task is to take level 1 radiance data from the satellites and to produce a level 2 (L2) product, called retrievals, such as vertical profiles and vertical columns.

AI/ML-driven retrieval algorithms have emerged as a powerful alternative to improve the processing efficiency of atmospheric composition satellite products. Physics-based retrievals, which are based on radiative transfer models (RTMs) and solves their inverse problem, have been widely used to generate satellite L2 products of atmospheric composition concentrations from observed spectral radiances. They consider detailed atmospheric processes to retrieve concentrations, but are computationally expensive. To speed up the retrieval processes, numerical inversion schemes have been replaced by ML algorithms that are trained using RTM inversions. Such an approach has been applied to satellite measurements to retrieve $O_3$ (e.g., (Müller et al., 2003)), $SO_2$ (e.g., (Li et al., 2022)), isoprene (e.g., (Wells et al., 2022)), and $CO_2$ (e.g., (Xie et al., 2024)).

In addition, ML techniques have been used to correct for satellite product bias and blending multiple products. For example, Oak et al. (2024) corrected the GEMS operational L2 $NO_2$ vertical column density with a ML model to match more mature TROPOMI observations, while preserving the GEMS data density. Similarly, Balasus et al. (2023) created a blend of TROPOMI+GOSAT methane products obtained by training the ML model to predict differences between TROPOMI and GOSAT co-located observations. Shi et al. (2024) developed an ozone column harmonization method using ConvNeXt (Liu et al., 2022b) to learn a mapping between OMI and TROPOMI, creating a reconstructed ozone column product with the long length of OMI availability and high spatial resolution and accuracy characteristics of TROPOMI. Such bias correction and blending approaches are powerful for providing accurate and consistent datasets for various science applications, for example, emission inversion.

#### 4.2.2 Fill in gaps in observational information

ML can also be used to fill gaps in observational information, such as supplementing missing data due to clouds to provide a continuous spatiotemporal distribution, and providing surface quantities that cannot be directly measured by satellite. Satellite observations of ozone and its precursors, combined with additional information such as meteorological conditions, land-use, population density, and anthropogenic emission inventories, have been used in NN or RF models to estimate spatiotemporal patterns of surface ozone concentrations at high spatial resolutions in different regions of the world (Di et al., 2017; Wang et al.,



2022; Zhu et al., 2022; Kang et al., 2021; Ghahremanloo et al., 2023). They contribute to air quality monitoring and human health impact assessment, with some studies focusing particularly on estimating health-relevant metrics using remote sensing data.

Di et al. (2017) proposed a hybrid neural network model using satellite-based data from OMI, GEOS-Chem CTM outputs, ozone vertical profiles, meteorological variables, land-use terms and other atmospheric compounds to predict MDA8 ozone in the continental United States. eXtreme Gradient Boosting (XGBoost) was used by Liu et al. (2020) to predict MDA8 ozone with similar inputs, while Jung et al. (2024) used XGBoost with OMI and MODIS products to estimate MDA8 at 1km resolution in Taiwan. Ghahremanloo et al. (2023) used a CNN with TROPOMI data as an input to estimate MDA8 in the United States.

Among various ML techniques, Zong et al. (2024) concluded that Deep Forest performs better than other shallower, tree-based regression models to estimate surface ozone from satellite ozone products. Similar surface concentration estimations based on NN or RF models have been applied to satellite $NO_2$ products to estimate surface $NO_2$ concentrations with high spatial resolution ((Kim et al., 2021), and to satellite aerosol optical depth measurements to estimate surface $PM_{2.5}$ concentrations (Huang et al., 2021; Xiao et al., 2021) which are useful for exposure estimates.

Emissions estimation using satellite observations of atmospheric composition concentration is another important ML application. ML techniques have been applied to improve the computational efficiency and accuracy of emissions estimation at various scales compared to traditional approaches based on data assimilation and other approaches (Dadheech et al., 2024; Xing et al., 2022; Tu et al., 2023; Li et al., 2024; Bruno et al., 2024).

In addition, AI-based anomaly detection methods pinpoint pollution hotspots, such as urban centers and areas of high indus-
trial activity. For instance, Joyce et al. (2023) developed a deep neural network to identify and quantify point source emissions of methane from hyperspectral images from the PRecursore IperSpettrale della Missione Applicativa (PRISMA) satellite with 30 m spatial resolution. ML models can also identify contributions from various emission sources (e.g., traffic, industry, wildfires) (Kang and Im, 2024; Finch et al., 2022; Kurchaba et al., 2023; Rollend et al., 2023).

ML can also be used to characterize key chemical environments and classify each area into different chemical regimes
based on satellite observations of pollutants and their precursors. For example, the abundance of the hydroxyl radical (OH) in urban areas initiates the removal of pollutants, making it a key species to describe the urban chemical environment. Despite its importance, it cannot be measured at the regional scale due to its very short chemical lifetime (Duncan et al., 2024). Zhu et al. (2022) developed a ML approach as an efficient alternative to computationally expensive CTM-based OH simulations, using observations of tropospheric $NO_2$ and HCHO columns from OMI as input to a ML model to predict daily near-surface
OH. Anderson et al. (2023) used a combination of ML models, CTM simulations, and various satellite observations to estimate tropospheric column OH in the tropics over the open oceans. They also showed that systematic biases in satellite products such as $NO_2$ may lead to biases in OH estimation.

These results indicate that combining satellite observations with ML approaches can provide important information for understanding and improving air pollution, including surface ozone and its precursor emissions, which cannot be directly mea-
sured from satellite observations. Further progress in this area can be expected through careful evaluation and understanding of




the characteristics and quality of satellite products, selection of effective supplementary information, and further development of appropriate ML methods.

## 4.3 Future Outlook and Priorities

Satellite observations are available at regional and global scales, for temporal scales ranging from hourly to decadal scales. As with the integration of ML with CTMs, priorities in the area of ML using satellite observations need to be further focused on their appropriate application at various spatio-temporal scales. With increasing resolution of satellite observations, the generalization of methods for producing satellite retrievals by ML has also become one of the priorities of the community.

Each satellite measurements have different characteristics, and data quality may vary not only spatially but also temporally. In many areas, especially in developing countries where air pollution is severe, ground truth observations are not available, hindering the application and validation of ML models. Careful validation of ML results is desirable, for example, taking into account the differences in spatial representativeness between satellite and ground-based observations. Another important challenge is how to integrate and use the results of validation using independent observations and uncertainty information from satellite retrievals as input for ML.

As the satellite constellation continue to evolve, data fusion of multiple instruments and multiple species using ML, in addition to chemical reanalysis approaches using data assimilation techniques (Inness et al., 2019; Miyazaki et al., 2020a), will play an important role in achieving consistent analysis of long-term trends. Combining satellite data with ML offers a further approach to investigate the linear and non-linear processes controlling air pollutant dynamics, complementing ground-based measurements and CTM estimates.

## 5 Challenges and Limitations

In this section, we reflect on some common challenges and limitations of using AI in the context of ozone forecasting, modeling and observations (Figure 4).

### 5.1 The challenges of data availability and workflow

Datasets are critically important to ML/AI research, and along with model choice and training methods, i.e. workflow, central to the success of modeling efforts and to their utility. As noted above, in the field of air pollution and atmospheric composition research, the use of ML is hampered by the absence of benchmark datasets suitable for training different model types with varying sizes and complexity. Such well-defined benchmarks including datasets, training objectives, evaluation scores, and baseline models have been instrumental for the rapid development of ML models in other fields (Dueben et al., 2022). In particular, WeatherBench and WeatherBench2 (Rasp et al., 2020, 2024), have been a key factor driving the transformation of ML weather forecasting between 2022 and 2024. Regarding surface ozone data, differences in the spatial coverage of ozone or of its precursors data will affect model development and the performance of data-hungry ML models. Ultimately, a lack of sufficient surface observations will impact the study of air quality and downstream impacts on health or vegetation. Therefore



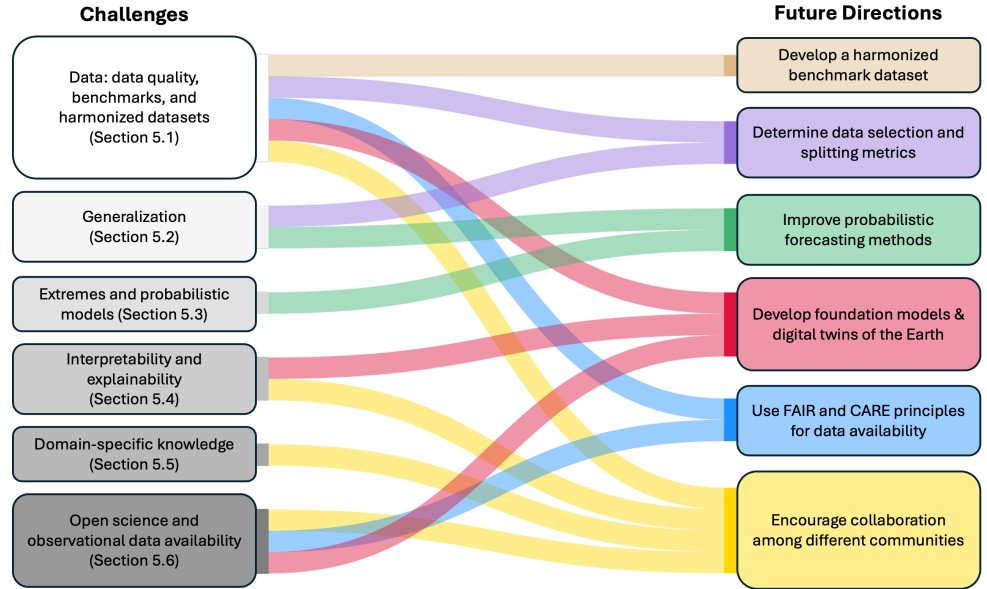

**Figure 4.** Challenges and future directions described in Sections 5-6. The left hand side represents the categories of challenges described in further detail in the sections listed. The right hand side lists future directions and priorities for AI/ML ozone studies. The lines connect the challenges with considerations for future work on the right hand side.

improving data coverage over poorly monitored areas remains a priority (Schultz et al., 2017). In contrast, the breadth of information available in datasets like TOAR, GHOST and reanalysis products like CAMS and TC is vast, but these products provide significant challenges to the development of ML models for ozone due to heterogeneous data formats and lack of
450  succinct documentation that focuses on the use of such data for ML applications. It may be that AI methods can also be used for infilling missing data (e.g. cloud-filtered satellite data, gaps in in-situ observations) for meteorological variables (Li et al., 2023) and ozone (Arroyo et al., 2018; Betancourt et al., 2022). Overall, there is a clear need for a harmonized benchmark dataset(s) for ozone to further enable ML models to be developed. These should (as much as possible) follow the vision outlined by Ebert-Uphoff et al. (2017) and the principles defined by Dueben et al. (2022).

455      With regards to model choice and development, it is worth noting that, in contrast to CTMs and other methods of simulation, ML models do not *a priori* require simulation, outputting and aggregating of high-resolution time series data to generate predictions for relevant ozone metrics. Instead, ML models can be trained to directly generate forecasts of these metrics (see section 2). In this regard, it is necessary that ozone ML benchmarks should, where appropriate, include target objectives both for forecasting concentrations and for forecasting (a set of) aggregate ozone metrics. See Fleming et al. (2018) and Lefohn
460  et al. (2018) for a more detailed discussion on relevant ozone metrics.

     With regard to model training, there is a wide array of data-splitting approaches that can answer subtly related scientific questions. As the motivating prediction challenges use both structured data and have structured prediction tasks, it is important





to match the way data are split to the scientific task at hand. For forecasting it is common practice to divide the data temporally such that the training data completely precedes the testing data, such as using the last few years of a longitudinal dataset for testing. This is commonly recommended in benchmarking studies (Lam et al., 2023; Rasp et al., 2020). We emphasize that while these procedures likely make intuitive sense, they do not match the default setting in ML packages (Schultz et al., 2021), such as scikit-learn (Pedregosa et al., 2011), where the default cross-validation procedure will randomly split over individual data instances rather than over spatial blocks or temporal blocks. Without using these correct procedures, performance will be overestimated and may not reflect real performance when deployed in practice. These challenges of data selection and splitting become particularly relevant when looking into climate timescales. Not only can this cause out-of-distribution samples of model input data (for example higher temperatures), but climate change may also affect atmospheric chemical and physical processes so that the mapping between inputs and outputs may drift. This problem is known as concept drift and some solutions have been developed in other ML applications. However, in practice, it remains a challenge to carefully define and document the data selection and splitting procedures and adapt them to the scientific problem at hand.

## 5.2    The challenge of generalization

In addition to appropriate handling of training data, ensuring the trained model is as generally useful as possible, both in and out of sample, remains an enduring challenge. Many ML models, particularly those trained as supervised learning problems, act as high-dimensional statistical interpolators. Consequently, these models may not genuinely learn the physics and chemistry of the atmospheric system but instead, learn a mapping from inputs to outputs based on the bounds of the training data. One significant challenge faced by these ML models, whether in supervised or unsupervised settings, is concept drift. Concept drift refers to ML predictions becoming less accurate over time, which can arise from a non-representative training dataset (initial conditions are narrow, do not sample enough timescales) or an ML model that lacks expressiveness, for example by being unable to extrapolate effectively beyond the bounds of the training data. For the latter, tree-based ML models especially are poor with respect to extremes and outliers. Here model architecture may play a role. Exploring generative AI models, such as Generative Adversarial Networks (GANs) and transformer models, holds promise for the next generation of ML-based atmospheric models. These newer ML architectures can generate more internally consistent dynamics and require less training data than classical CNNs when fine-tuned from previously well-trained models, while also demonstrating improved accuracy and stability over time.

As is common in ML tasks, models trained on data from one geographical region may not necessarily transfer to another region, even when the underlying task and physics remain the same. This limitation often arises from variations in spurious features or unobserved variables specific to each domain. For example, if a model is trained on data from one country with specific emissions and climate, it may not be expected that this model will perform as accurately in another country with a very different distribution of data. Many approaches in the ML literature seek to improve the performance of ML models across domains, or under domain shifts, which are yet to be used for ozone forecasting (Sagawa et al., 2019), while recent studies suggest that large-scale weather forecasting models may generalise to unseen conditions and perturbations (Hakim and Masanam, 2024). Generalisation is particularly important for the use of ML models trained in high-data domains and then



deployed in low-data domains. In this context, it may be useful to exploit the benefits of probabilistic forecasting, using models that report uncertainty in unfamiliar domains.

To what extent these results would apply to large-scale ML models for ozone remains to be seen. For tropospheric ozone, the correctness of capturing atmospheric chemical processes that are not always directly observable, but which can have an influence on ozone concentrations, particularly on longer time scales, may be important. Examples of this include HONO, nitrate and aldehyde formation and conversion processes, which provide for "storage of radicals" during nighttime. These radicals can be released with sunrise in the morning hours and drastically increase ozone formation rate. While it may be possible that sufficiently complex ML models capture enough auxiliary information to model this ozone boost correctly, such case studies will become important and have not yet been explored.

## 5.3 The challenges of extremes and probabilistic models

A relevant application for ozone forecasting and modeling is the study of extremes, including both accurate forecasting and attribution. Extreme ozone concentrations or fluxes can have a large impact on health and vegetation, and are also referred to as low-likelihood high-impact (LLHI) events. By definition, extreme events occur rarely and are hence challenging to accurately represent. There has been some work on approaches to weight extremes more during model training (Steininger et al., 2021). The ability of models to represent extremes is also an important metric that can be used to evaluate the quality of these models. Extremes can thus play a role for uncertainty quantification of the predictive performance of the models (e.g., important for forecast emulators and assessing ML performance on the extremes), where one can distinguish between epistemic (systematic) and aleatoric (statistical) performance. This connects with an increasingly recognized need to evaluate performance in more rigorous and consistent ways: including the development of new benchmark datasets, diagnostics and metrics (see above). Progress on ML evaluation include causal evaluation (process-oriented approach) and eXplainable AI (xAI), for understanding (in)consistencies of the ML algorithms with physical processes (in other words: whether accurate answers are found for the right reasons). However, such methods are generally only applicable to relatively small-scale ML models. Dynamical tests and counterfactual experiments provide a means to test the credibility of large ML models (Hakim and Masanam, 2024; Baño-Medina et al., 2024).

Forecasting of potential extreme events is particularly challenging because these events are beyond the typical ozone variability. Naturally, extreme events are rarely and infrequently represented in data, and therefore supervised ML approaches trained with non-extreme data may fail to predict extremes well. For data-driven models this challenge is further exacerbated by (1) the need to forecast not only the presence of threshold exceedances, but also the intensity and duration of extreme events; and (2) ozone extreme events are often related to other anomalous mechanisms, such as heatwaves and wildfires, which are difficult to take into account based on limited extreme information, also due to the fact that ozone responses to these mechanisms are heterogeneous. Although the extreme value theory is widely adopted, its limitations are frequently acknowledged, including the IID (independent and identically distributed) assumption and independence between extreme and non-extreme events. On the other hand, approaches based on probabilistic forecasting may better characterize the uncertainty and likelihood of extreme events. One solution may be to use metrics and data scenarios to evaluate performance under different types of





evaluation scenarios, taking advantage of evaluation metrics in weather forecasting which have been studied extensively. For example, if a key consideration is the ability of a forecasting model to capture extreme events, then metrics that capture relevant performance explicitly on those events should be used. This allows for robust comparison of both the existing and novel models on both traditional metrics and metrics focused on extreme event prediction to more comprehensively evaluate model

performance. Evaluation of extreme events is limited in the literature, with recent studies highlighting the lower accuracy of ML models when forecasting spring and summertime ozone concentrations (Leufen et al., 2023; Hickman et al., 2023).

In addition to helping with forecasting extreme events, probabilistic forecasting more generally provides a number of advantages compared to the deterministic forecasting methods that are currently more common (Bodnar et al., 2024). Well calibrated forecast uncertainty is important for delivering accurate air quality warnings and for public information. Furthermore for other

tasks where data quality is critical, such as infilling missing data for use in later analysis, well-calibrated estimates of uncertainty provide information on the quality of ML generated data and hence on further analysis. This is especially relevant as many infilling approaches use average values, and therefore may not be applicable for extremes.

## 5.4    The challenge of interpretability and explainability

Interpreting and explaining ML models used to study ozone remains difficult. While these two terms are often used inter-

changeably, for this article we follow the distinction that interpretability focuses on designing and exploring models that are transparent and the internal data transformations are comprehensible to us, whereas explainability methods focus on post-hoc explanations of how black-box models are working (Rudin, 2019). Models that are directly and trivially interpretable, such as multiple linear regression, are typically not the most performant, and in the high-data regime, the most performant ML models are typically variants on deep neural networks that are difficult to interpret or explain. There is some literature that explores

whether PINNs provide more interpretability. For example, efforts are underway to enhance the interpretability of ML models in atmospheric sciences by incorporating or diagnosing conservation priorities such as mass and stoichiometry (Sturm and Wexler, 2020, 2022). Additionally, neural operators are being employed to learn the solution operators of ODEs/PDEs from the chemical training data (Liu et al., 2024). However, while incorporating chemistry and physics constraints has been shown to increase interpretability, there is no guarantee that these methods will improve the stability of the ML model over time (Sturm

et al., 2023). Often, there is a trade-off between interpretability and ML model accuracy, especially with more complex models (Sengupta et al., 2023).

While methods to interpret and explain neural networks more generally have been studied widely, mechanistic interpretability of neural networks is a challenging task (Nanda et al., 2023), and only a limited range of XAI methods have been tested with ML methods developed for ozone forecasting, often focused on sensitivity approaches which look at the post-hoc expla-

nations where the inputs to models are perturbed to see how predictions change (Ivanovs et al., 2021). Recent studies have investigated the importance of model input parameters through bootstrapping, i.e. random perturbations of individual inputs (Kleinert et al., 2021). Input data perturbation experiments are also possible and informative for very large models as, for example, demonstrated by Hakim and Masanam (2024) for the Pangu-Weather forecast model.





## 5.5 The challenges arising from domain-specific knowledge

Modeling ozone using ML proves challenging due to the multitude of sources driving model error (emissions, chemistry, transport, deposition, representativeness) and the nonlinear response of ozone to these sources. Parameter tuning an appropriate ozone ML model for a complex, high-dimensional parameter space is possible given large computational resources and adaptive learning on pre-defined metrics. However, such an approach is largely inefficient given that atmospheric chemistry data lies on relatively low-dimensional manifolds with respect to the possible input parameter space. That is, many ozone-related

relationships are structured with individual signals often being sparse and low-rank. Here, domain knowledge from atmospheric chemistry can help identify the optimal training dataset and define meaningful loss functions and targeted timescales (Figure 1) for the ML model problem. For example, ML models of atmospheric chemistry tend to predict well fast chemical processes (e.g., seconds to days) but diverge over longer time scales (e.g., months to years) (Kelp et al., 2020). Knowledge of slow chemical processes, such as the role of peroxyacetyl nitrate (PAN) decomposition for ozone formation over polluted

and/or remote areas, may help define appropriate training targets for ML models. An emphasis should be placed on emulating chemistry on longer timescales (> 1 year) as issues of long-range stability are more challenging than shorter-term accuracy, and are a necessity for inclusion into CCMs and ESMs.

Alternatively, a heightened focus on domain knowledge may unintentionally limit the potential of ML models. Atmospheric chemists typically leverage well-established relationships of the chemical system, such as $NO_X$-limited vs. VOC-limited

regimes, which are easily uncovered by linear regression or principal component analysis. By invoking such a strong prior assumption, it may impose constraints that hinder an ML model's ability to learn more complex, non-obvious interactions within the data. This bias toward known relationships, while useful for capturing dominant signals, risks overlooking patterns that could be hidden in the chemical state space that may promote greater accuracy and stability over longer time scales. While these uncovered relationships may not always represent true causal mechanisms, the value of ML lies in its capacity to dis-

cover interactions that may escape human intuition. Striking a balance between leveraging domain expertise and allowing ML models the flexibility to explore complex dynamics is essential for advancing the predictive capability of ozone modeling.

## 5.6 The challenges of Open Science and Observational Data Availability

To thrive, the interdisciplinary ozone modeling and forecasting community requires open knowledge sharing, open resources and research cooperation. Research in the domain should adhere to the FAIR principles of Findability, Accessibility, Interop-

erability and Reusability (Wilkinson et al., 2008) and the CARE principles of Collective, Authority to control, Responsibility and Ethics (Stephanie Russo Carroll et al., 2021). Availability of data is essential for data-driven approaches and the developed TOAR-II surface ozone database is key here through its open data policies and its Application Programming Interface (API), which allow for automatic extraction of data and integration into a workflow (e.g. in Jupyter Notebooks). Besides the aforementioned "FAIR data", a call for FAIR-software has been recognized and expressed more recently (Barker et al., 2022), thus

including a larger part of the research cycle than only data collection, storage and dissemination.





Although an open data infrastructure such as the TOAR-II database gives the impression of low barriers to data access, this might in fact not be true for everyone. Poor internet connectivity from developing countries may limit researchers from retrieving data and subsequently running a computationally demanding model (Blanken et al., 2022; Dwivedi et al., 2022). The increasing resolution of satellite products and models is often considered to be an improvement, but the larger data size can

complicate the processing and analysis of data for some researchers (Jain et al., 2022). The concept of data availability may thus be perceived differently across regions. There are also additional barriers accessing data. Some data services require a registration and compliance with data use policies, which could conflict with institutional policies of researchers or exacerbate language barriers that non-native English researchers can experience. Finally, whereas advanced APIs can be ideal for technically skilled researchers and allow for reproducible workflows, they might hinder less technical researchers or policy makers

that want to explore data sets.

In particular to developing nations, which may not have the economic ability to acquire high-resolution satellite products outside of those freely-available, it is imperative to develop high-quality, globally generalizable solutions to ozone modeling. Data hosting platforms like Google Earth Engine (GEE) enable users to freely access relevant, global data relevant for ozone modeling studies, ranging from land-use information from MODIS (Friedl, 2021) to human modification data from

VIIRS nighttime lights (Elvidge et al., 2017), Gridded Population of the World (University, 2018), and more. Recent work by Kazemi Garajeh et al. (2023) investigated the ability to detect spatially resolved ozone pollution trends using time-series Sentinel-5 imagery from GEE, highlighting the quality of spatial distribution and accuracy available an open-source product and platform. This demonstrates the necessity to co-design data services and their hosting platforms to provide efficient and performant access to high-quality, well-documented data.

## 6 Future Directions

ML advances are swiftly revolutionizing atmospheric composition research in general, and ozone thermodynamics and chemical processing in particular, harnessing both Earth observations and modeling outputs to extract valuable insights from vast datasets.

A wealth of Earth observation data from satellites, in-situ measurements, and ground-based instruments is now available,

with data volumes already well beyond tens of petabytes. This abundance of data is marking a golden age for Earth observation. New developments in deep learning (DL) are rapidly overcoming limitations related to spatio-temporal and multi-variable relationships, promising significant breakthroughs (Eyring et al., 2024). However, their direct application to Earth observations is often hindered by challenges such as data quality and coverage, model generalization across different regions, and the complexity of associated metadata, such as relative position, orientation, and shading in satellite imagery. Future research and

advancements in ML-based observational products, including efforts to address uncertainty quantification e.g. (Haynes et al., 2023), will further enhance our understanding, facilitate process-based model evaluation (Nowack et al., 2020), and enable actionable science, particularly in areas such as human health (Fleming et al., 2018) and climate impact (Gaudel et al., 2018) assessments.



Accurate forecasts of extrema in short-term surface $O_3$ predictions are essential for protecting human health, while reliable
projections of long-term changes in tropospheric $O_3$ abundances are critical for understanding climate change and avoiding its
most severe impacts. Leveraging causal- and physics-constrained data-driven approaches can enhance trust and interpretability
in ML-based modeling efforts (Tesch et al., 2023; Beucler et al., 2024). Combining causal discovery and eXplainable Artificial
Intelligence (XAI) methods holds potential for advanced process-based evaluation (Iglesias-Suarez et al., 2024). This approach
can enable the identification of physical and chemical processes that contribute to both good and poor model performance,
providing valuable insights into the drivers of ozone, including sources and sinks. On the other hand, these methods are
generally limited to small-scale ML models and different approaches (e.g. analysis of attention maps) need to be sought for
large-scale ML models. There is a recognized need to evaluate model performance rigorously and consistently, calling for the
development of new benchmark datasets, diagnostics, and metrics (Betancourt et al., 2021, 2022), to enable comprehensive
evaluation and validation of ML-based ozone modeling techniques. Yet, current domain-specific ML methods and tailored
solutions often lack a more universal perspective, whereas general-purpose ML-based algorithms would only need to be fine-
tuned to address specific tasks.

Nevertheless, within the context of ML-based solutions for atmospheric composition, including ozone, and numerical mod-
eling, future work can focus on foundation models to advance more integrated approaches. Foundation models, as large-scale,
pre-trained architectures, are emerging as promising tools for atmospheric modeling. These models, built on extensive datasets
in a self-supervised manner Bommasani et al. (2021), have already demonstrated their adaptability in fields like weather fore-
casting and climate science (Lessig et al., 2023; Nguyen et al., 2023; Bodnar et al., 2024). In the context of tropospheric
ozone modeling, foundation models could reduce computational burdens while learning from varied datasets, including obser-
vational and numerical modeling data (Mukkavilli et al., 2023). These models are capable of handling multiple air pollutants
simultaneously and can incorporate meteorological variables, supporting the development of more comprehensive, flexible and
potentially robust air quality benchmarks by supplementing and harmonizing observational data. Their flexible architecture
enables training a single model with large-scale resources and fine-tuning it for multiple tasks, reducing the computational ex-
pense of repeated model training (Bommasani et al., 2021). For ozone forecasting, they offer a path toward improved modeling
capabilities by leveraging diverse data sources and capturing complex relationships in atmospheric composition. Recent work
has shown that foundation model architectures can simulate atmospheric dynamics and composition by emulating variables
from reanalysis products like CAMS (Inness et al., 2019), achieving promising results for multi-day air pollutants forecasting,
including ozone (Bodnar et al., 2024). This shift would, in turn, facilitate the development of models that are not only specific
to ozone but capable of generalizing to broader air quality applications, potentially leading to more robust and versatile ML
solutions.

For CTMs, particularly, and ESMs more broadly, foundation models can serve as valuable tools to either complement or
enhance traditional approaches. Chemistry schemes and parameterizations based on ML models can bring computationally
efficient interactive chemistry to ESMs for accurate long-term projections (Eyring et al., 2016). Additionally, end-to-end em-
ulators can complement physics-based simulations by employing large ensembles to better constrain uncertainties. While
general-purpose ML algorithms, such as foundation models, can provide a step change in data-driven approaches, they may



not always lead to actionable science, such as contributing to more informed decision-making by, for example, accurately attributing ozone pollution events to specific sources— and the development of effective mitigation strategies.

The need for actionable science to address challenges like climate change and air quality has led to the development of digital twins of the Earth (DTEs), which integrate ML models and physical simulations to provide comprehensive monitoring and forecasting of the Earth system (Bauer et al., 2021; Bauer, 2024). DTEs enable users to explore "what-if" scenarios and develop targeted strategies, such as mitigating ozone pollution and improving air quality, by simulating the impacts of various interventions. This user-centric approach contrasts with traditional top-down methods and supports more effective, adaptive responses to environmental challenges. However, the implementation of DTEs raises critical issues regarding governance, data accessibility, and computational demands, particularly as they rely on exascale computing to deliver real-time insights (Hazeleger et al., 2024). As global initiatives like the European Commission's Destination Earth and NASA's Earth System Digital Twin advance these frameworks, ML techniques will be essential for ensuring their efficiency and scalability, enabling DTEs to serve as a transformative tool for improving air quality and addressing broader environmental challenges.

To meet society's needs facing current environmental challenges by providing actionable science and maintaining the rapid progress in this field, collaboration among different communities is essential. The full potential of ML in atmospheric composition in particular, and Earth system science in general, can only be realized through an interdisciplinary approach, fostering close cooperation between domain-knowledge and ML communities. Furthermore, this requires opening new collaborative opportunities between academia and the private sector. As awareness grows within the ML community regarding the societal relevance of algorithms in Earth system research, major technology firms are increasingly interested in applying their expertise to environmental-related issues. They may well pursue this through interdisciplinary research collaborations with domain-expert scientists, whether through direct employment or academic partnerships. As these applications extend into unexplored future climates, input from academic experts will become increasingly crucial, facilitating collaborative efforts to advance the understanding of the Earth system.

## 7 Conclusions

While modeling ozone accurately remains a challenging problem across temporal and spatial scales, ML approaches have made progress in a number of areas. As highlighted in this Perspective, ML methods are contributing to research in short-term forecasting, chemical mechanism emulation and remote sensing of ozone. Specifically, ML methods are providing increasingly accurate short-term forecasts of ozone at observational stations, and making progress toward providing fast emulators of chemical mechanisms used in chemistry-climate models. In remote sensing, ML methods have shown skill in increasing the efficiency of ozone retrieval, and in making estimates of ozone where there is little satellite coverage.

In addition to excelling at specific tasks, recent work illustrates that foundation models, trained on diverse datasets, are capable of more general atmospheric composition modeling, including ozone. This paradigm of foundation models represents a significant step forward for composition modeling, enabling an integrated approach across multiple scales and tasks, and building on the success of similar models for weather forecasting. For this work to make further progress in modeling real-





world ozone faithfully, models should be trained with a synthesis of high-quality observational datasets. In order to facilitate robust, rapid progress in ML for ozone modeling, appropriate high-quality benchmarks must be compiled to evaluate the skill of different models, and enable the comparison of ML and numerical models. Furthermore, continued work to mitigate the

challenges faced by existing ML models is necessary, which will require close collaboration between domain experts and ML researchers to develop models tailored to the specific challenges of ozone modeling. Finally, it remains an open and important question whether ML models can contribute to improved process-level understanding of the drivers of ozone, and generalise to unseen air pollution and climate scenarios.

As ML continues to transform ozone research and adjacent fields, including weather and climate modeling, the ozone

modeling community needs to ensure future research builds on strong foundations. By developing robust benchmarks, building meaningful cross-disciplinary collaborations, and embracing state-of-the-art techniques, ML-driven ozone research has the potential to not only advance scientific understanding but also deliver actionable benefits for climate resilience and public health.

## Appendix A: Glossary

**Artificial Intelligence (AI):**
A software or model that is capable of performing tasks that typically require human intelligence.

**Copernicus Atmosphere Monitoring Service (CAMS):**
A service by the EU Earth observation programme to provide comprehensive data on atmospheric composition and air quality through satellite and ground-based monitoring.

**Chemistry-Climate Model (CCM):**
A type of global model focused on the interactions between atmospheric chemistry and climate.

**Convolutional Neural Network (CNN):**
A type of neural network designed for processing data with grid structure, often used for image processing.

**Chemical Transport Model (CTM):**
A type of global model designed to simulate the movement and chemical reactions of atmospheric pollutants.

**Deep Forest (DF):**
A deep learning architecture based on decision trees instead of neural networks.

**Deep Learning (DL):**
A field of machine learning focused on the development and use of neural networks.

**Decision Tree (DT):**
A hierarchical supervised learning algorithm, often used to create classification and regression models.



**Earth System Model (ESM):**

> A global model that simulates all aspects of the Earth system, including the interactions between the atmosphere, oceans, land, and biosphere.

**Feed-forward Neural Network (FNN):**

> A basic type of neural network where data move in one direction without feedback loops, often used for data classification and recognition.

**Foundation Model (FM):**

> A machine learning model trained on vast amounts of data, designed to be adapted to a broad range of tasks.

**Generative Adversarial Network (GAN):**

> A type of machine learning technique where two neural networks compete unsupervised to produce the most accurate result.

**General Circulation Model (GCM):**

> A global model that simulates the Earth's atmospheric dynamics and circulation.

**Gradient Boosted Decision Tree (GBDT):**

> An ensemble machine learning technique that uses the results of multiple decision trees to improve accuracy and reduce error of the prediction.

**Large Language Model (LLM):**

> A type of foundation model trained on very large text datasets to understand and generate natural language.

**Long Short-Term Memory network (LSTM):**

> A type of recurrent neural network designed to retain information over longer sequences for longer periods.

**Machine Learning (ML):**

> A field of artificial intelligence dedicated to algorithms and models that can learn and make predictions from the input data without being explicitly programmed to do so.

**Neural Network (NN):**

> A machine learning model designed to process data in a similar way as the human brain.

**Physics-Informed Neural Network (PINN):**

> A type of neural network trained to follow known physical laws.

**Random Forest (RF):**

> An ensemble machine learning algorithm that combines multiple decision trees during the training process to improve prediction accuracy and reduce overfitting.



**Recurrent Neural Networks (RNN):**

A type of neural network in which data can loop back into the network retaining memory of previous inputs. It is designed for sequential data processing where context is important, such as natural language and time series.

**Transformer Model (TM):**

A type of deep learning model that converts a given input into a desired output, learning context and meaning. It is used as a foundation model for large language models as an alternative to CNNs and RNNs architectures.

**U-Net:**

A type of convolutional neural network designed for image segmentation and de-noising.

**eXplainable AI (xAI):**

A type of artificial intelligence that provides the information necessary to understand how a certain output was achieved.

*Code and data availability.*  All code and plotting routines are available at (ML4O3, 2024).

*Author contributions.*  All authors contributed to the writing and review of the manuscript. SHMH, MK, PTG, F-IS, GK, KD, KZ, MGS and EAP led the writing and preparation of the MS. SHMH led Section 2, MK section 3, KD, EAP and KM led section 4, GK section 5 and FI-S
section 6.

*Competing interests.*  S.H.M.H., M.K., D.E.C., P.T.G. declare no competing interests. Some of the authors are involved with editorial work for Copernicus journals.

*Acknowledgements.*  S.H.M.H. acknowledges funding from EPSRC via the AI4ER CDT at the University of Cambridge (EP/S022961/1). P.T.G. acknowledges the Environmental Geochemical Cycle Research Group, Earth Surface System Research Center, Japan Agency For
Marine Science and Technology for support as an External Researcher. P.T.G. and A.T.A. were financially supported by NERC through NCAS (R8/H12/83/003). Part of this work was conducted at the Jet Propulsion Laboratory, California Institute of Technology, under contract with the NASA. We acknowledge the support of the National Aeronautics and Space Administration (NASA) Atmospheric Composition: Aura Science Team Program (19-AURAST19-0044), Atmospheric Composition Modeling and Analysis Program (22-ACMAP22-0013), NASA Earth Science U.S. Participating Investigator program (22-EUSPI22-0005). Z.L. was acknowledges the National Natural Science
Foundation of China, grant number 42307140. MGS acknowledges funding by the European Commission under grant ERC-AdvG-787576. K.-L.C. was supported by NOAA cooperative agreement (no. NA22OAR4320151). J.J.W. acknowledges National Aeronautics and Space Administration (NASA) grants no. NNX16AQ30G and no. 80NSSC23K0930. D.E.C. acknowledges financial support from Underwriters Laboratories, Inc.





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
