# Peer review of "Applications of Machine Learning and Artificial Intelligence in Tropospheric Ozone Research"

_EGUsphere, 2024_

## Author Comment (AC1)

We thank the reviewers for their comments on this review. We address the points in turn.

*This is a comprehensive, well-written review written by experts in the field, and clearly deserves to be published in GMD. It will be useful for the customary purposes of a review paper (e.g. giving active participants in the reviewed field and in closely related fields entry points into the literature, providing nice summary graphics, and discussing methodologies and challenges that will underlie future research). As someone working on ML applications in a related geophysical field, I find that most of the broader themes in the text (e.g. heterogeneous datasets, end-to-end prediction, issues and challenges having to do with learning wide ranges of space and time scales and lots of correlated predictors, long-term emulator drift, explainability and PINNs, effective benchmarks and intercomparisons, foundation models) would apply just as well to my own area of research.*

We thank the reviewer for their positive comments, and appreciate their care in reviewing the Perspective.

*One thing I look for in a review article is to highlight some crisp, intellectually exciting problems that could launch a new student or postdoc into career-launching research directions. One could glean inspirations from the 'Future Outlook' subsections and Section 5 on 'Challenges and Limitations' and 'Future Directions', but the issues raised there mostly involve large coordinated efforts with a heavy software engineering focus. One could argue that such efforts are the primary path to further progress in ML for tropospheric ozone and related chemistry, but are there also relevant conceptual questions you'd like to highlight that are more accessible to academic researchers?*

Thank you for this helpful suggestion. In the revised manuscript, we expand Figure 4 to highlight some tangible next steps and include how they connect to the Challenges and Future Directions identified, and we also include new text (L554-568) highlighting some accessible and topical research questions.

*Specific comments*

*L181: Reference formatting*

*L199: Delete 'so'*

*L243: What is an 'NMB'?*

*L386: What is 'MDA8'?*

Thank you for noting these points. These have now been addressed.

*R2: General Comments*

*The authors provide a perspective piece on the use of machine learning/artificial intelligence (ML/AI) to help with various aspects of prediction and process understanding surrounding atmospheric ozone (O3) concentrations. The authors discuss the chemical processes and scales that control O3 concentrations and then they lay out the various reasons why O3 concentrations are important for human health and other impacts (primarily in the troposphere), and discuss the challenges in the current state of observing and predicting those concentrations. They then focus on three aspects of ML/AI application towards O3 prediction: 1) making short-term predictions for specific ground locations related to health hazard thresholds and operational forecasting programs, 2) predicting O3 within regional and global models, which is currently possible though computationally demanding via chemical transport modules within atmospheric dynamic models, and 3) improving how remotely-sensed information about O3 and its related chemical species can be incorporated into production datasets and forecasts. For each of these sections they discuss recent research and challenges around ML/AI. They end with broad identification of unsolved challenges and possible paths forward for making better use of ML/AI in this field.*

*Overall, I found the paper to be very comprehensive and well written, and it was useful for me as someone who is not an expert on O3, but who is generally familiar with the challenges of using ML/AI in physical modeling, to see where progress has been made in this particular field. I think that some of the authors' recommendations, such as the need for a benchmark dataset that the O3 modeling community can agree upon to drive progress in ML/AI forecasts, will be helpful to organize progress.*

We are very grateful to the reviewer for their time and these comments.

*I found that the framing of the paper is at times unclear, as many of the issues brought up are not specific to O3 modeling, but are instead general to ML/AI and/or its application to physical modeling. This meant that it was often unclear whether the challenges being discussed really were specific to O3 and if not, whether they really are the most important items to mention in this context. Relatedly, the manuscript often repeats general issues in several locations, making it overly long.*

Thank you for the comment. We have made several modifications made to streamline the paper and remove repetition:

1. The last few paragraphs of the Introduction were condensed to briefly outline the paper without reiterating the points that are made in subsequent sections.
2. The "Future Outlook and Priorities" subsections of each of Sections 2, 3 and 4 have been removed, with issues raised in these subsections now mentioned in Sections 5 and 6 to shorten the manuscript and limit repetition. In addition, some remaining paragraphs in Sections 2-4 have been shortened.
3. Many of the topics in Section 5.2 were repetitive with those in Sections 5.1 and 5.3, so Section 5.2 was significantly shortened.

4. Details from Section 6 were removed and replaced with shorter summary paragraphs. Specific information from the Section 5 subsections was moved to Section 6 instead, and recommendations for tangible, ozone-specific challenges were added to Section 6.

*Regarding specific sections:*

*Section 3.2: I was a bit unclear on the difference between 3.2.1 (ML emulation and reduced order modeling) and 3.2.2 (ML models implemented within global CTMs). Are the studies in the former section essentially doing "offline" ML emulation of model output datasets? And the latter section is "online"? For example, "reduced order modeling" could seem to apply to either situation.*

The sections have been renamed "3.2.1 Offline ML and reduced order modeling" and "3.2.2 Online ML within global models."

*Section 3.3: One challenge that I did not see discussed in the paper, and would be appropriate to mention here, is how to marry the current success in ML/AI weather and climate model emulation with chemical transport modeling. Thus far ML/AI emulators have largely excluded CTMs from their scope. The current framework of training these emulators also generally is non-extensible, for example it is not possible to add even a conservative tracer to the ML framework that already has been trained on atmospheric dynamics, even though conceptually the ML should be able to "know" about tracer advection. Instead the entire training must be rerun with the tracer among the predicted variables. Do the authors see any approaches for bridging this gap? It could dramatically speed up the inclusion of species like O3 in the predictions made by ML/AI emulators.*

We have added a paragraph to the 'Future Directions' section to address the reviewer's viewpoints:

"At present, there are underexplored opportunities to merge the current successes in ML weather and climate model emulation with CTMs and ESMs. Thus far, atmospheric chemistry data have been largely excluded from ML weather and climate applications, as these current supervised learning frameworks are typically non-extensible, requiring retraining of the entire ML model when incorporating new chemical information. In contrast, unsupervised learning model frameworks, such as pre-trained foundation models, can identify patterns in data without explicit labels, offering a new frontier for ingesting and potentially improving ML modeling of atmospheric chemistry. These foundation models can be fine-tuned on CTM data. For example, the Aurora model (Bodnar et al., 2025) is fine-tuned on a subset of six criteria pollutants, including ozone from CAMS (Inness et al., 2019). Fine-tuning ML weather and climate models enables the addition of chemical species to an ML model that is already trained on atmospheric dynamics. This process of fine-tuning, by training specific decoders for new variables, has also recently been carried out for hydrological variables (Lehmann et al., 2025). However, adding chemical species such as VOCs in the absence of emission inputs (which current models do not consider) on the ML weather model's native 6-hour forecast time steps

likely presents challenges. Greater emphasis is needed on understanding the factors influencing ML model performance with respect to the specific challenges of atmospheric composition research and air quality analysis."

*Section 3.3 It is also worth noting that the ML/AI weather forecasting state of the art has moved heavily towards probabilistic/diffusion-based architecture that have the inherent ability to produce sharp forecasts and uncertainty estimates. How does this impact potential for O3 forecasts?*

We have added text in 5.3 to address this viewpoint:

"Furthermore, ML weather forecasting models are increasingly adopting probabilistic and diffusion-based architectures that are able to produce sharp forecasts and uncertainty estimates. This is a promising line of work, however, these ML architectures may be challenging to implement for ozone forecasts due to the uncertainty driven by the meteorological fields themselves."

*Sections 5 and 6: I found that these sections repeated points made early rather heavily and could have been more concise.*

Thank you for this helpful comment. As noted above, we have shortened both Sections 5 and 6, and have removed the "Future Outlook and Priorities" subsections from Chapters 2-4 to avoid repetition of points, which we hope improves the readability of the paper.

*I am also curious if the authors are willing to offer opinions regarding which of the recommendations they find most likely to produce success for the modeling community in the short and medium term. For example, the manuscript lists many possible paths forward, but are all equally important? It seems likely that a benchmark dataset could spur advances in ML/AI skill for O3, and the authors list this as the first suggestion. Are the other suggestions equally likely to produce success, or are they more general suggestions for good practices or things that would be "nice to have"? For example, while XAI and PINNs offer conceptual and process understanding benefits, they lack a track record of success in ML/AI forecasts as compared to deep learning black box approaches. How do the authors feel that these goals should be prioritized?*

Thank you for raising this point. To address this comment to some extent, we have emphasised some accessible research projects which we believe should be prioritised in the final paragraph of Section 6. While we do not give an explicit prioritisation/ranking of research problems, we hope this addition, and rephrasing in parts of Sections 5 and 6 addresses this comment.

*Specific Comments*

*Figure 2: Some explanation of the meaning of the arrows and colors in the lower panel would be helpful.*

Thank you for noting this. We have now added a further description of the arrows and the colors in the caption, as follows.

Lower panel: long-term ozone trends based on monthly anomalies at remote surface sites. Red and blue indicate positive or negative trends respectively, with different shades giving the statistical significance of the trend at each site.

*TOAR is not properly introduced on first reference, for those not familiar with the acronym*

Thank you, this is now done.